

# 1 Climate changes in interior semi-arid
# 2 Spain from the last interglacial to the
# 3 late Holocene

Dongyang Wei[1,2], Penélope González-Sampériz[3], Graciela Gil-Romera[3], Sandy P.
Harrison[1], I. Colin Prentice[4]
1: Department of Geography and Environmental Science, University of Reading,
Whiteknights, Reading, RG6 6AB, UK
2: Masters Programme in Ecosystem and Environmental Change, Department of Life
Sciences, Imperial College London, Ascot, SL5 7PY, UK
3: Instituto Pirenaico de Ecología-CSIC, Avda. Montañana 1005, 50059 Zaragoza,
Spain
4: AXA Chair Programme in Biosphere and Climate Impacts, Department of Life
Sciences, Imperial College London, Ascot, SL5 7PY, UK





**Abstract**

The El Cañizar de Villarquemado sequence provides a palaeoenvironmental record from the western Mediterranean Basin spanning the interval from the last part of MIS6 to the late Holocene. The pollen and sedimentological records provide qualitative information about changes in temperature seasonality and moisture conditions. We use Weighted Averaging Partial Least-Squares (WA-PLS) regression to derive quantitative reconstructions of winter and summer temperature regimes from the pollen data, expressed in terms of the mean temperature of the coldest month (MTCO) and growing degree days above a baseline of 0° C ($GDD_0$) respectively. We also reconstruct a moisture index (MI), the ratio of annual precipitation to annual potential evapotranspiration, taking account of the effect of low $CO_2$ on water use efficiency. We find a rapid summer warming at the transition to MIS5. Summers were cold during MIS4 and MIS2, but some intervals in MIS3 were characterized by summers as warm as the warmest phases of MIS5 or the Holocene. However, MIS3 was not significantly warmer in winter than other intervals, and there was a gradual decline in winter temperature from MIS4 through MIS3 to MIS2. The pronounced changes in temperature seasonality during MIS5 and MIS1 are consistent with changes in summer insolation. The ecophysiological effects of changing $CO_2$ concentration through the glacial cycle has a significant impact on reconstructed MI. Conditions became progressively more humid during MIS5 and MIS4 was also relatively humid, while MIS3 was more arid. High MI values are reconstructed during the deglaciation and there was a pronounced increase in aridity during the Holocene. Changes in MI are anti-correlated with changes in $GDD_0$, with increased MI during intervals of summer warming indicating a strong influence of temperature on evapotranspiration. Although our main focus here is on longterm changes in climate, the Villarquemado record also shows millennial-scale changes corresponding to Dansgaard-Oeschger cycles.



## 1 Introduction

The modern climate of the western Mediterranean region is influenced by high pressure
systems in summer and westerly storm tracks in winter, giving rise to a highly seasonal
climate with dry summers, wetter winters and strong seasonal temperature contrasts.
The region is sensitive to both extratropical and low-latitude influences, and therefore
registers climate changes on glacial-interglacial time scales due to changes in ice sheet
volume and on orbital time scales due to changes in insolation (Magri and Tzedakis,
2000; Rohling et al., 2013). Records from the western Mediterranean region also show
abrupt climate changes during the glacial associated with the Dansgaard-Oeschger (D-
O) cycles and Heinrich events (Sánchez-Goñi et al., 2002; Fletcher et al., 2010; Vegas
et al., 2010; Moreno et al., 2012). In addition, continental records from the Iberian
peninsula show abrupt changes during the late glacial and early Holocene (e.g.
González-Sampériz et al., 2006; Moreno et al., 2012; Pérez-Sanz et al., 2013; Ramos-
Román et al., 2018).

The El Cañizar de Villarquemado palaeolake (hereafter Villarquemado) is located in
the semi-arid interior of the Iberian Peninsula (40.49°N, 1.29°W, 985 m a.s.l.), and
provides a long, continuous pollen record that stretches from the end of the penultimate
glaciation (Marine Isotope Stage 6, MIS6) through the last interglacial (MIS5), the last
glaciation (MIS4, 3 and 2) and into MIS1 and the Holocene (Moreno et al., 2012;
González-Sampériz et al., 2013; Aranbarri et al., 2014). The record has 30 radiocarbon,
IRSL and OSL dates and a robust independent chronology constructed using Bayesian
modelling (Valero-Garcés et al.). The length of the record and the quality of the age
model make Villarquemado a uniquely important site to understand climate changes in
Spain on both glacial-interglacial and orbital timescales. The only other record from
Spain that spans this length of time is Padul (Pons and Reille, 1988; Camuera et al.,
2018), in southern Spain. The chronology for the interval before 50 ka at Padul is based
on amino-acid racemization and an assumption of constant sedimentation rates and is
thus poorly constrained compared to the Villarquemado record. Most other long records
from the Mediterranean region, including the classic site of Tenaghi Phillippon
(Tzedakis et al., 2006; Milner et al., 2013) and the newer record from Lake Ohrid
(Wagner et al., 2017; Sinopoli et al., 2018; Sinopoli et al., 2019), use some form of
orbital tuning in constructing an age model. The only exception is Lago di Monticchio
in central Italy (Allen et al., 2000; Allen and Huntley, 2009; Martin-Puertas et al., 2014;
Allen and Huntley, 2018) – but while this has a highly resolved and independent
chronology, the site is in a much wetter climate today than Villarquemado. Interglacial
intervals in Lago di Monticchio are dominated by temperate deciduous trees whereas
Villarquemado is characterized by evergreen trees and shrubs (González-Sampériz et
al., 2013).

In this paper, we present a quantitative reconstruction of three bioclimatic variables:
winter temperature (mean temperature of the coldest month, MTCO), growing-season
warmth (growing degree days above a base level of 0°C, $GDD_0$) and a moisture index
(the ratio of annual precipitation to annual potential evapotranspiration, MI) using the
pollen record from Villarquemado and Weighted Averaging Partial Least-Squares
regression (WA-PLS: ter Braak and Juggins, 1993). We apply a novel method of
correcting MI to take account of the direct physiological influence of $[CO_2]$ on water
use efficiency. Although Villarquemado is well dated, there are several intervals with
poor pollen preservation, so we focus on the long-term evolution of climate rather than
the evidence for abrupt climate changes. Finally, we compare our reconstructions with
available pollen-based reconstructions from the wider Mediterranean region, and





discuss the implications of the reconstructed changes.
**2 Methods**
**2.1 Modern pollen data**
The modern pollen dataset (Fig. 1) consists of records from 6458 terrestrial sites. The
bulk of the sites were derived from the European Modern Pollen Database (EMPD) v
3.0 (Davis et al., 2013) and the EMBSeCBIO (Eastern Mediterranean-Black Sea-
Caspian corridor BIOmes) Initiative (Marinova et al., 2017). We included additional
sites from various publications (Saadi and Bernard, 1991; de Klerk et al., 2009; Gruger
and Jerz, 2010; Muller et al, 2010; Werner et al., 2010; Tarasov et al, 2011; Matthias et
al., 2015; Niemeyer et al., 2015; Bell & Fletcher, 2016; Novenko et al., 2017) available
from the European Pollen Database (http://www.europeanpollendatabase.net/) or
Pangaea (https://www.pangaea.de/). We also included long-term pollen trap data. In
addition, we included 73 modern surface samples from northern Spain (see
Supplementary Information Table 1, and González-Sampériz, 1999; Garcia-Prieto,
2015; Aranbarri et al., 2015; Leunda et al., 2017; Rieradevall et al., 2018).
We standardised the taxonomy for the pollen data using Plants of the World Online
(www.plantsoftheworldonline.org/) and the Integrated Taxonomic Information System
(https://www.itis.gov/). We removed obligate aquatics (e.g. *Azolla*, *Lemna*,
*Myriophyllum*), insectivorous plants (e.g. *Drosera*), parasitic plants (e.g. *Bartsia*) and
introduced species (e.g. *Eucalyptus*, *Liquidambar*) assuming that the distribution of
these plants is only partially related to climate. We also removed cultivated plants (e.g.
*Avena*, *Cannabis*, *Hordeum*), although we retained taxa (e.g. *Olea, Prunus*) that can be
cultivated but also occur in the wild. Even after these deletions, there are still many
(1558) taxa in the modern data set. However, it cannot be assumed that all taxa have
been identified consistently to the same level of taxonomic discrimination. A further
problem is that discrimination of sub-types is unnecessary in regions where only one
sub-type is present (e.g. it is not usually stated whether *Quercus* is deciduous or
evergreen in northern Europe, where all the *Quercus* species present are deciduous).
Some families are only represented by a single genus in the data set (e.g. *Verbena* is the
only representative of the Verbenaceae, most other genera in this family are tropical);
in such cases, preserving both family and genus is meaningless. Finally, although some
herbaceous species are recognizable at species level, they do not occur in distinctive
climate regimes, and thus preservation of these as individual species does not convey
additional information about climate. We therefore reduced the original taxon list by
amalgamating taxa into a manageable list. The amalgamation process was guided by
palynological and ecological understanding of the pollen types and tested by
constructing climate space diagrams for each original and amalgamated taxon using
generalised additive models (GAMs).
The GAMs were implemented with the *mgcv* R package (Wood, 2017). The R
implementation makes the selection of the smoothing parameters automatic (Guisan et
al., 2002). We used a square root transformation of MI, as differences between MI
values at the wet end are less important than differences at the dry end in terms of their
effect on vegetation (Prentice et al., 2017). Logistic models were used in the first step
of the GAMs. The fitted response surfaces show the concentration of the pollen taxon
abundance in climate space. Interaction terms were not included, because we assume
that each bioclimate variable independently influences the distribution of plant taxa and



vegetation types, following the logic of Wang et al. (2013). For visualization purposes,
the 3D response surfaces of taxon abundance resulting from the GAM were portrayed
as slices for low, medium and high values of $GDD_0$. Convex hulls, implemented with
the *alphahull* and *ggplot2* packages in R (Pateiro-Lopez & Rodriguez-Casal, 2016,
Wickham, 2016), were used to show the area where samples actually lie and thus avoid
representing parts of the fitted surface that were not closely constrained by data.
Supplementary Table 2 provides the translation of the original taxa into the taxon list
used in our analyses. The final taxon list (249 taxa) includes several layers of specificity:
individual species, genera, sub-families, and families. All pollen data were transformed
from raw counts to relative abundance prior to analysis. Amalgamated taxa that occur
in less than 10 sites were not considered in the final analysis, which therefore only uses
taxa.
**2.2 Modern climate data and derivation of bioclimatic variables**
Climatological data (mean monthly temperature, precipitation, and fractional sunshine
hours) were derived from the CRU CL v2.0 gridded dataset of modern (1961-1990)
surface climate at 10 arc minute resolution (~18 km) (New et al., 2002). Geographically
weighted regression (GWR) was carried out in ArcGIS (v10.3, ESRI, 2014) to correct
for elevation differences between each pollen site and the corresponding grid cell. A
fixed bandwidth kernel of 1.06° (~140km) was used because this optimized model
diagnostics and reduced spatial clustering of residuals relative to other bandwidths. The
climate of each pollen site was then estimated based on its longitude, latitude, and
elevation. The mean temperature of the coldest month (MTCO) was taken directly from
the GWR regression. Growing degree days above 0°C ($GDD_0$) were estimated from
daily data using a mean-conserving interpolation (Rymes and Myers, 2001) of the
monthly mean temperatures. The annual Moisture Index, defined as the ratio of annual
precipitation to annual potential evapotranspiration (MI), was calculated for each pollen
site using code modified from SPLASH v1.0 (Davis et al., 2017) based on daily values
of precipitation, temperature and sunshine hours again obtained using a mean-
conserving interpolation of the monthly values of each.
**2.3 Statistical analyses**
Canonical correspondence analysis (CCA; ter Braak, 1986; Legendre & Legendre,
2012) was used to perform a constrained ordination of the modern pollen data in
response to the bioclimatic variables. CCA was implemented with the *vegan* package
(Oksanen et al., 2017) in R (v3.3.1). We excluded predictors with variance inflation
factors (VIFs), which give a measure of the multi-collinearity in predictors, higher than
20. The significance of the CCA model was computed with an ANOVA-like
permutation test. The CCA allows an assessment of the degree to which the bioclimate
variables reflect the main pattern of variability in the modern pollen data (Table 1).
**2.4 Fossil pollen data**
The Villarquemado palaeolake (González-Sampériz et al., 2013) is located in the Jiloca
basin in the semi-arid region of north-eastern Spain (Fig. 1). The site is occupied today
by a wetland and cultivated land. The surrounding vegetation is dominated by evergreen
trees (*Quercus ilex*, *Q. coccifera*, *Q. faginea*) and xerophytic shrubs (e.g. *Rhamnus*
*lycioides*, *Genista scorpius*, *Ephedra fragilis*, *Thymus vulgaris*, *T. zygis*). *Juniperus* (*J.*





*thurifera*, *J. communis*, *J. sabina*, *J. oxycedrus*) and *Pinus* (*P. sylvestris*, *P. pinaster*)
occur at higher elevations. A 74m-long core, taken from the deepest part of the wetland,
provides a pollen and sedimentological record back to MIS6 (Moreno et al., 2012;
González-Sampériz et al, 2013; Aranbarri et al., 2014; Blas Valero-Garcés, submitted).
The Bayesian age model is based on 30 $^{14}$C, IRSL and OSL dates. The age model was
constructed using BACON v2.2 (Blaauw and Christen, 2011). Full details of the age
model are given in Valero-Garcés et al. Sedimentation rates are low during the initial
part of the record and increase from the beginning of MIS2 onwards. There are intervals
with poor pollen preservation between 16 085 and 22 328, 31 203 and 37 482, 43 112
and 50 103, and 87 895 and 93 809 cal yr BP. The average pollen sampling interval is
ca 300 yr, increasing to ca 140 yr during MIS1. In general at least 300 pollen grains
were counted per sample; no sample has less than 150 grains counted.
**2. 5 WA-PLS**
The modern bioclimatic and pollen data were used to develop pollen-climate transfer
functions independently for MTCO, $GDD_0$ and MI using weighted-averaging partial
least squares regression (WA-PLS) (ter Braak and Juggins, 1993). Like CCA, WA-PLS
is based on the assumption that each taxon has a unimodal distribution in climate space.
It is relatively robust to spatial autocorrelation, and uses model residuals to diminish
bias and improve performance. WA-PLS was implemented with the *rioja* R package
(Juggins, 2017). The performance of the calibration models was assessed through leave-
one-out cross validation. The number of components used in each model was estimated
through a randomisation *t*-test on the results (Van der Voet, 1994). We selected the
component with the lowest root mean square error of prediction (RMSEP), but only if
there was a significant improvement in RMSEP relative to a lower number of
components – since including more components can result in over-fitting of the data so
that model predictive value decreases (ter Braak et al., 1993). We checked that the final
transfer functions had a high $R^2$ for prediction and a low maximum bias.
**2.6 Correcting for changing [$CO_2$] concentration**
In addition to affecting plants indirectly through changes in climate, atmospheric $CO_2$
concentration [$CO_2$] has a direct effect on plant physiological processes (Ehleringer et
al., 1997; Farquhar, 1997; Prentice and Harrison, 2009). Increasing [$CO_2$] allows plants
that use the standard $C_3$ pathway of photosynthesis (including temperate grasses and
forbs, and nearly all trees) to assimilate more carbon while losing less water, implying
an increase in water use efficiency (Bramley et al., 2013). Under conditions of low
[$CO_2$], $C_3$ plants are less productive and this can also result in a shift in the balance of
$C_3$ and $C_4$ plants. Pollen-based reconstructions that rely on calibration of pollen
abundance against modern climate values do not account for the direct effects low [$CO_2$]
on water use efficiency, and as a result reconstructions of moisture variables, such as
precipitation and MI, register drier conditions than actually occurred (Prentice et al.,
2017). Prentice et al. (2017) have developed a procedure to correct for this, and we have
applied this correction using the implementation described in Cleator et al. (submitted).
The procedure requires the specification of [$CO_2$] and mean annual temperature (MAT)
at the site. We used the ice-core [$CO_2$] record (Bereiter et al., 2015), using a loess
smoothing spline with a span of 0.1. We calculated MAT from the reconstructed MTCO
and $GDD_0$ at a site (see Appendix 1). This calculation also allowed us to generate an
estimate of the mean temperature of the warmest month (MTWA), which can then be
used to generate a time series of temperature seasonality (MTWA−MTCO). We applied
the correction to the downcore reconstructions of MI at Villarquemado.





**3 Results**
The CCA analysis shows a strong correlation between species abundance and the three
climate variables in the modern pollen data set, with correlations of 0.83, 0.61 and 0.47
respectively (Table 1). The VIF scores for each bioclimatic variable are low (<6),
indicating that they are reasonably independent, and the CCA shows that they each have
an independent contribution to explaining the variation in abundance. This is confirmed
by the ANOVA-like permutation test, which shows that the bioclimatic variables and
the three variability axes are all significantly different from one another (Table 1).
For the construction of the WA-PLS regression, we used results from component 4 for
MTCO and $GDD_0$ and component 3 for MI because these are the significant results
with the lowest RMSE and highest $R^2$. The $R^2$ values are 0.69, 0.66 and 0.52 for MTCO,
$GDD_0$ and MI respectively (Supplementary Table 3). Nevertheless, close examination
of the downcore reconstructions showed there were anomalous peaks in reconstructed
MI, particularly at the end of MIS5. These correspond to samples that have unusually
high values of Poaceae and Polypodiales (Fig. 2), and where the sedimentary record
indicates that these are likely to be aquatics. Both Poaceae and Polypodiales were
therefore removed from the final WA-PLS model (Table 2). This made no change to
the number of components or the goodness-of-fit of the model, but made the
reconstructions of MI for the anomalous samples less extreme and more plausible (Fig.
3). It had no significant impact on the MTCO and $GDD_0$ reconstructions
(Supplementary Fig. 1). We checked whether particularly high or low values in the
temperature reconstructions were a result of anomalous characteristics of the pollen
assemblages, specifically whether there was evidence of pollen degradation (as
measured by the number of indeterminable grains in the sample) or the samples were
characterized by low biodiversity (as measured by the N2 index: Hill, 1973). There was
no evidence of a correlation between the abundance of indeterminable grains and
anomalously high or low reconstruction values. However, depauperate samples tended
to produce more extreme temperature values than adjacent more diverse sample
(Supplementary Fig. 2). We therefore exclude samples with an N2 value <2 from the
final reconstructions. Excluding samples with a value of <3 had little effect on the
reconstructions but increased the patchiness of the reconstructions by removing a large
number of samples.
The reconstructions (Fig. 4, Supplementary Table 4) show an increase in both summer
($GDD_0$) and winter (MTCO) temperature between 130 and 127 ka. There is a general
trend for both summer and winter temperature to decline through MIS5, and although
there are fluctuations, they do not correspond exactly with the chronological boundaries
of sub-stages within MIS5 (Fig. 4; definition of stage and sub-stage boundaries in the
Supplementary Table 5). Furthermore, the changes in summer and winter temperature
are not in phase. Minimum winter temperatures occurred earlier than minimum summer
temperature in MIS5e, so that winter temperatures were already increasing while
summer temperatures continued to decrease after ca 120 ka. In contrast, during MIS5a,
warming in summer occurred at broadly the same time as winter cooling such that the
temperature seasonality was significantly enhanced between ca 80 and 70 ka. There
was no pronounced cooling, either in summer or winter, at the transition into the glacial
(Fig. 4). The record from both MIS3 and MIS 2 is not continuous and the available
samples may show the response to millennial-scale events; thus it is difficult to
characterise the general trends in temperature. However, MIS 3 appears to have been





somewhat warmer than both MIS2 and MIS4 in summer. MIS3 was not significantly
warmer in winter than other intervals during the glacial, and indeed there was a gradual
decline in winter temperature from MIS4 through MIS3 to MIS2. MIS 1 was
characterized by a general warming trend in both summer and winter, although the
reconstructions show considerable variability superimposed on this trend.

The implied increase in temperature seasonality during MIS5e, 5c, 5a and during the
early part of MIS1 corresponds to increased seasonality in insolation compared to the
present day (Fig. 5), primarily driven by high summer insolation. Insolation changes
across the glacial were comparatively muted. Intervals of increased temperature
seasonality during MIS3, therefore, cannot be explained by changes in the seasonality
of insolation.

Conditions became progressively more humid from MIS5e through to MIS5c, while
conditions were generally humid but variable during MIS5a (Fig. 4, Table 3). MIS4
was also relatively humid, while MIS3 was the most arid phase reconstructed during
the glacial. However, the difference in reconstructed MI between MIS4 or MIS2 and
MIS3 is not large. This reflects the fact that $[CO_2]$ decreased throughout the glacial, so
that the impact of the $CO_2$ correction becomes larger from MIS4 through MIS3 and into
MIS2 (Supplementary Fig. 3). The influence of changing $[CO_2]$ is most marked in
comparatively dry climates (Fig. 6), which is why this effect has such an important
influence on the reconstructions of MI at Villarquemado. The highest values of MI are
reconstructed during the deglaciation and there was a pronounced increase in aridity
during the Holocene. The reconstructed changes in MI are broadly anti-correlated with
changes in $GDD_0$ ($r = -0.69$), with decreased MI during intervals of summer warming.
This suggests that the changes in MI were largely driven by changes in
evapotranspiration rather than changes in precipitation.

There are several abrupt changes shown in the reconstructions, most particularly during
MIS5a and in the glacial period. Some of these (Supplementary Fig. 4) clearly
correspond to D-O events, including D-O 20 (72.28-70.28 cal ka) and 19 (76.4-74 cal
ka) in MIS5a and 9 (40.11-39.81 cal ka) and 8 (38.17-36.57 cal ka) in MIS3. Heinrich
Stadial 2 (26.45-24.25 cal ka) also clearly corresponds to an interval of year-round
cooling in our reconstructions. Gaps in the pollen record, and poor dating resolution in
some parts of the record, preclude identification of all of the D-O and Heinrich events.
However, where D-O events are registered, they were characterized by a marked
increase in seasonality – this explains the apparently anomalous high seasonality
recorded during some parts of the glacial (Fig. 5).

## 4 Discussion and Conclusion

The Villarquemado record is characterized by rapid warming in winter temperature of
ca 5°C and an increase in the summer growing season of ca 2000 degree days over a
period of ca 2-3 kyr during the transitions from MIS6 to MIS5e. Although there are
fluctuations, there is an overall decline in both summer and winter temperature through
MIS5. However, there is a major interval with poor pollen preservation during MIS5b,
and this limits our ability to gain a complete picture of the evolution of climate during
this interval. Temperatures reconstructions for MIS4, 3 and 2 do not appear to be



significantly lower than the end of MIS5, but this may be because the coldest intervals occur during the intervals of low pollen preservation in MIS 3 and 2. The Younger Dryas interval is marked by relatively cold summers. There was a gradual warming in both summer and winter through the Holocene. The broad-scale changes in moisture are in general coherent with changes in $GDD_0$, with warmer summer intervals characterised by drier conditions and colder summers by wetter conditions. However, the MI reconstructions indicate that the whole of the past ca 130 kyr was wetter than today. Rapid millennial-scale changes in temperature and moisture are superimposed on these longer-term trends, though not all D-O events can be identified in the Villarquemado record.

Many of the features of the Villaquemado record are shown in other quantitative reconstructions from the Mediterranean. Both the Monticchio (Allen et al., 2002; Allen and Huntley, 2009) and the Lake Ohrid (Sinopoli et al., 2019) record show rapid warming at the transition from MIS6 to MIS5e. This warming occurs over longer period in the Monticchio (ca 5kyr) and Ohrid (ca 7 kyr) records. Differences between the sub-stages of MIS5 are more pronounced in the Monticchio and Ohrid records than at Villarquemado. The comparison of modern analogue and WA-PLS reconstructions at Lake Ohrid shows that modern analogue reconstructions (and by implication the response-surface approach used at Monticchio) tend to produce stronger fluctuations, and this might contribute to explaining the more muted variability at Villarquemado. However, the pronounced cold, dry interval registered in Monticchio and Ohrid during 5b corresponds to an interval of low pollen preservation in Villarquemado. This, and the fact that the Villarquemado site lies at the warmer and drier end of the climate gradient across the Mediterranean, could contribute to the apparent differences between the sites.

The coldest interval at Monticchio during MIS 2 is also represented by a hiatus in Villarquemado. The Younger Dryas was characterised by cooler summers and wetter conditions at Villarquemado, but only a small decrease in winter temperature. The wetter conditions and the muted winter temperature response are consistent with the record from Monticchio (Allen and Huntley, 2009). However, the Holocene record from Monticchio is very different from the pattern of climate change shown at Villarquemado. Whereas the Villarquemado record is characterised by warming and drying, the Monticchio record shows summer cooling and relatively stable moisture levels after 10 ka. Thus, while there are some similarities between the available quantitative records from the Mediterranean, they each show distinctive features reflecting the complexity of climate changes across the region and differences in modern climate and vegetation. This complexity will only be resolved when more quantitative reconstructions, preferably using a consistent methodology, are available from the circum-Mediterranean region.

The temperature record at Villarquemado shows intervals of enhanced seasonality during MIS5 and MIS1, largely but not entirely driven by changes in $GDD_0$ (a reflection of summer temperature and the length of the growing season). We have shown that there is a good correlation between these intervals of enhanced seasonality and orbitally-forced changes in summer insolation. Insolation changes across the glacial were comparatively small and this is reflected in muted changes in temperature seasonality. Orbital forcing was not the only cause of enhanced seasonality at Villarquemado, since we also see enhanced seasonality during D-O events (e.g. D-O 9). However, enhanced seasonality during the D-O events appears to have been driven primarily by changes in winter temperature. On both orbital and millennial time scales,





changes in MI are generally anti-correlated with changes in $GDD_0$ presumably because increased summer temperature and/or increased length of the growing season led to increased evapotranspiration and hence reduced MI.

There are major gaps in the palynological record from Villarquemado because of intervals of poor pollen preservation during MIS5b, MIS3 and MIS2 (Fig. 2). The sedimentological record suggests that these were arid intervals, characterized by alluvial fan rather than lacustrine deposition, and oxidation of the sediments. A speleothem record from El Pindal (Moreno et al., 2010) also show hiatuses in formation during MIS2, consistent with our interpretation that the depositional hiatus at Villarquemado is indicative of pronounced aridity. Similar situations have been identified in other palynological sequences during arid intervals (Valero-Garcés et al., 2000, 2004; Vegas-Villarubía et al., 2013; González-Sampériz et al., 2004, 2005). Hyper-arid periods are a problem for pollen preservation and, while further work may improve the pollen record at Villarquemado, it is unlikely that we will be able to obtain a quantitative record of climate during such intervals. Nevertheless, Villarquemado provides the most complete and well-dated record from continental Iberia (González-Sampériz et al., 2010; Moreno et al., 2012) and it is important to document changes in the drier part of the circum-Mediterranean region.

All reconstruction methods that use modern pollen-climate relationships are sensitive to the choice of training data sets, vegetation diversity and the potential absence of analogue assemblages (Gavin et al., 2003; Jackson and Williams, 2004; Bartlein et al., 2011). However, the training data set that we have used contains more than 6000 samples and includes samples from very cold and very warm environments to allow reconstruction of climate both much warmer and much colder than today. Analysis of the GAMs for individual taxa also shows that they have ecologically plausible relationships with climate variables. We have shown that pollen preservation issues, as indicated by intervals when the number of indeterminable grains was higher than average, do not affect our reconstructions. However, our analyses show that intervals of very low biodiversity are often characterized by more extreme values than other intervals. We have taken this into account by screening the down-core samples and excluding samples that have very low diversity.

Much of the discussion about the lack of modern analogues has focused on interpretation of assemblages of species that are not found together today (Overpeck et al., 1985; Jackson and Williams, 2004; Williams and Shuman, 2008). However, one important non-analogue situation that is ignored in all previous reconstructions is the impact of $[CO_2]$ different from today on plant assemblages. This does not affect temperature reconstructions but has a significant effect on moisture-related variables such as precipitation or any moisture index (Prentice et al., 2017; Cleator et al., submitted). Taking the impact of $[CO_2]$ into account in our reconstructions reduces the variability of MI during glacial intervals. Prentice et al. (2017) showed that this correction produced a reconciliation of apparently contradictory interpretations of pollen and geomorphic data for hydroclimatic changes in southeastern Australia at the Last Glacial Maximum. Comparison of our reconstructions from Villarquemado with other hydroclimatic data would be useful to test the realism of the reconstructed MI changes.

The Villarquemado reconstructions provide a detailed picture of the response of western Mediterranean climate and vegetation to changes in external forcing in this sensitive region for a long time. It would be useful to generate quantitative

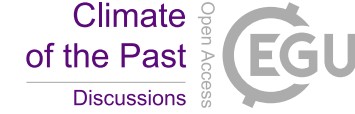



reconstructions from other long records, since preliminary comparisons with Lago di
Monticchio and Lake Ohrid indicate some complexity in the response of climate to
changes in external forcing within the circum-Mediterranean region. However, given
that the largest impact of glacial-interglacial changes in atmospheric circulation is likely
to be on precipitation and plant-available moisture, it will be important to take account
of the impact of changing $[CO_2]$ in these reconstructions.

**Acknowledgements**
DW and SPH acknowledge support from the ERC-funded project GC2.0 (Global
Change 2.0: Unlocking the past for a clearer future, grant no. 694481). PG-S, GG-R
and SPH acknowledge support from the Spanish Ministerio de Economia y
Competitividad for the project CGL2015-69160R "Dinámica, Monitorización y
Calibración de la vegetación Mediterránea en respuesta al Calentamiento Global en
series temporales largas (DINAMO3), as well as for CGL2012-33063 and CGL2009-
07992. ICP acknowledges support from the ERC-funded project Re-inventing
Ecosystem and Land-surface Models (REALM), grant no. 787203. This research is a
contribution to the AXA Chair Programme in Biosphere and Climate Impacts and the
Imperial College initiative on Grand Challenges in Ecosystems and the Environment
(ICP). We thank Jon Lloyd for advice on the implementation of convex hulls, and Maria
Dance for the implementation of GWR. We also thank the PaleoIPE team for multi-
proxy reconstruction and discussion regarding Villarquemado sequence; Maria Leunda,
Josu Aranbarri and Héctor Romanos for providing additional surface pollen samples;
and Miguel Sevilla Callejo for the Iberian Peninsula and Villarquemado vegetation
maps of Figure 1.



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



**Figures and Table Captions**

Figure 1. Location of modern pollen samples and fossil pollen site. (a) Map of the Iberian Peninsula showing the location of Villarquemado superimposed on a simplified elevation map of the region. (b) Vegetation map of Villarquemado area showing dominant tree species, surface area of the basin and location of the core site. (c) Map showing the location of Villarquemado and the distribution of modern pollen samples. The background map shows mean temperature of the coldest month (MTCO, °C).

Figure 2: Simplified stratigraphic and pollen diagram from Villarquemado. The first column shows changes in stratigraphy, including the alternation between lacustrine and non-lacustrine conditions. The simplified pollen diagram shows the changing abundance of Mediterranean and steppe taxa. We also show the changing abundance of Poaceae and Polypodiales. The final column shows the biodiversity index (Hill's N2).

Figure 3: The impact of removing Poaceae and Polypodiales from the taxon set on reconstructions of moisture index (the ratio of annual precipitation to annual potential evapotranspiration, MI) during MIS4 and MIS5a. The red line (without P/P) is the reconstructed values once Poaceae and Polypodiales are removed. Removing these two taxa reduces anomalous peaks, where they were particularly abundant, but has little impact on the reconstructions for the rest of the core.

Figure 4: Reconstructed mean temperature of the coldest month (MTCO, °C), growing degree days above a base level of 0° C ($GDD_0$) and moisture index (the ratio of annual precipitation to annual potential evapotranspiration, MI). Only samples with a Hill's N2 biodiversity index >2 are plotted. The Marine Isotope Stages (MIS) and substages are shown by vertical dotted lines and labelled; we also show the transition interval between MIS6 and MIS5e. Red dots indicate the modern climate calculated from the elevation-corrected climate data from the Climate Research Unit (CRU) CL 2.0 data set.

Figure 5: The correlation of the temperature seasonality and insolation. The black line in the top plot is the normalized difference of reconstructed mean temperature of the coldest month and the mean temperature of the warmest month calculated based on MTCO and $GDD_0$ (Appendix 1). The orange line in the top plot is the difference between July and January insolation in $W\ m^{-2}$ at 40.49 °N (the latitude of Villarquemado). The bottom panel shows mid-monthly insolation anomalies (compared to present) in $W\ m^{-2}$ at 40.49N through time.

Figure 6: Scatter plot showing the impact of the $[CO_2]$ correction on the reconstructed moisture index (MI). The coloured dots represent the implied change in the reconstructions, grouped according to level of the actual $[CO_2]$ at that time (in ppm).

Table 1: Summary statistics for the first three axes of CCA in the whole European data set. The analysis was based on 6458 sites, 196 taxa and three bioclimatic variables: mean temperature of the coldest month (MTCO, °C), growing degree days above 0° C ($GDD_0$) and the moisture index (MI). We also show the summary statistics for the ANOVA-like permutation test (999 permutations).

Table 2: The results of randomisation $t$-test on the leave-one-out cross-validated predictions of the weighted averaging-partial least squares (WA-PLS) regression





models used for the climate reconstructions. The final model is based on 194 taxa,
omitting Poaceae and Polypdiales.  Selected components in the final model are marked
in bold.
Table 3: Reconstructed average values of mean temperature of the coldest month
(MTCO, °C), growing degree days above a base level of 0° C ($GDD_0$) and moisture
index (the ratio of annual precipitation to annual potential evapotranspiration, MI) for
Marine Isotope Stages (MIS) and substages, calculated from the interpolated yearly
values of each variable.



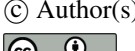

**Appendix 1**

**1. An approximation of the dependence of $GDD_0$ (growing degree days on a zero**
**base) on $T_{min}$ (coldest-month temperature) and $T_0$ (mean annual temperature)**
Assume that temperature (T) follows a sinusoidal pattern with time of year:
$T = T_0 - \Delta T \cos \theta$
where $\Delta T$ is the half-amplitude of seasonal variation in temperature, and $\theta$ is the 'day
angle':
$\theta = (2\pi/365) \, t_d$
where $t_d$ is the day of the year, measured from a starting point in midwinter. Then:
$T_{min} = T_0 - \Delta T$
$GDD_0 = \int_{T>0} (T_0 - \Delta T \cos \theta) \, d\theta$    (in units of K rad, where 1 K rad $= 2\pi/365$ K day)
The day-angle when T = 0 is given by $\theta_0 = \cos^{-1}(T_0/\Delta T)$, except in two special cases:
1) $T_0 > \Delta T => GDD_0 = 2\pi T_0$ (this is the case when $T_{min} \geq 0$, hence every day is a growing
day)
2) $-T_0 > \Delta T => GDD_0 = 0$ (this is the case when there are no growing days)
Otherwise:
$GDD_0 = 2 \left[ T_0 \theta - \Delta T \sin \theta \right]$ evaluated from $\theta_0$ to $\pi$
$= 2 \pi T_0 - 2 T_0 \cos^{-1}(T_0/\Delta T) - 2 \Delta T [\sin \pi - \sin \cos^{-1}(T_0/\Delta T)]$
$= 2 \pi T_0 - 2 T_0 \cos^{-1}(T_0/\Delta T) - 2 \Delta T \sqrt{[1 - (T_0/\Delta T)^2]}$
Write $u = T_0/\Delta T$
Then $GDD_0 = 2 \Delta T [\pi u - u \cos^{-1} u + \sqrt{(1 - u^2)}]$
$= 2 \Delta T [u \cos^{-1}(-u) + \sqrt{(1 - u^2)}]$.
This is in units of K rad. Multiplication by $365/2\pi$ converts this to units of K day.
**2. Predicting $T_0$ from $GDD_0$ and $T_{min}$**
From the logic above:
$-T_{min}/\Delta T = 1 - u$ *and* $GDD_0/\Delta T = 2 [u \cos^{-1}(-u) + \sqrt{(1 - u^2)}]$
Therefore:
$-GDD_0 / T_{min} = 2 [u \cos^{-1}(-u) + \sqrt{(1 - u^2)}] / (1 - u)$



$$= 2 \left\{ [u/(1-u)] \cos^{-1}(-u) + \sqrt{[(1+u)/(1-u)]} \right\}.$$
To estimate mean temperature ($T_0$): convert $GDD_0$ from K day to K rad, take the ratio
of $GDD_0$ to ($-T_{min}$), and solve the equation above for $u$. Then,
$T_0 = -T_{min}\, u/(1-u)$
and
$\Delta T = -T_{min}/(1-u).$



Figure 1. Location of modern pollen samples and fossil pollen site. (a) Map of the
Iberian Peninsula showing the location of Villarquemado superimposed on a simplified
elevation map of the region. (b) Vegetation map of Villarquemado area showing
dominant tree species, surface area of the basin and location of the core site. (c) Map
showing the location of Villarquemado and the distribution of modern pollen samples.
The background map shows mean temperature of the coldest month (MTCO, °C).

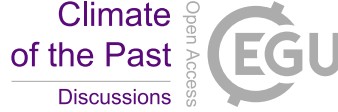



Figure 2: Simplified stratigraphic and pollen diagram from Villarquemado. The first
column shows changes in stratigraphy, including the alternation between lacustrine and
non-lacustrine conditions. The simplified pollen diagram shows the changing
abundance of Mediterranean and steppe taxa. We also show the changing abundance of
Poaceae and Polypodiales. The final column shows the biodiversity index (Hill's N2).

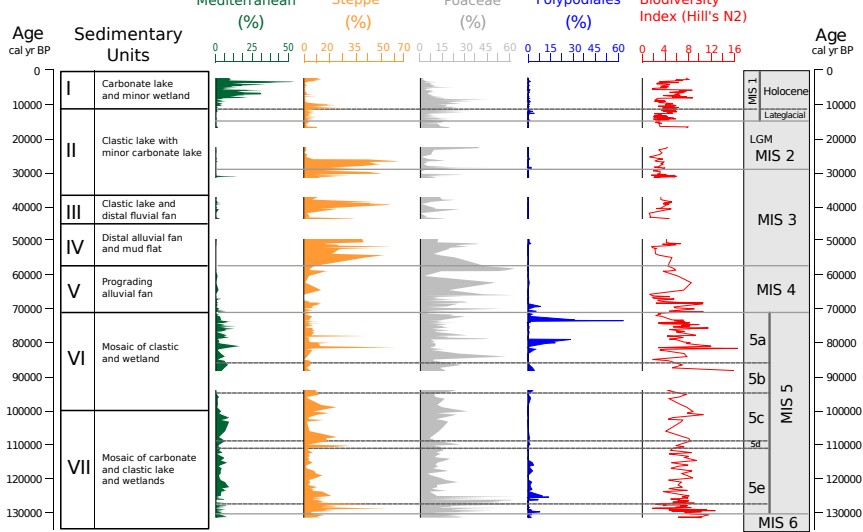






Figure 3: The impact of removing Poaceae and Polypodiales from the taxon set on
reconstructions of moisture index (the ratio of annual precipitation to annual potential
evapotranspiration, MI) during MIS4 and MIS5a. The red line (without P/P) is the
reconstructed values once Poaceae and Polypodiales are removed. Removing these two
taxa reduces anomalous peaks, where they were particularly abundant, but has little
impact on the reconstructions for the rest of the core.

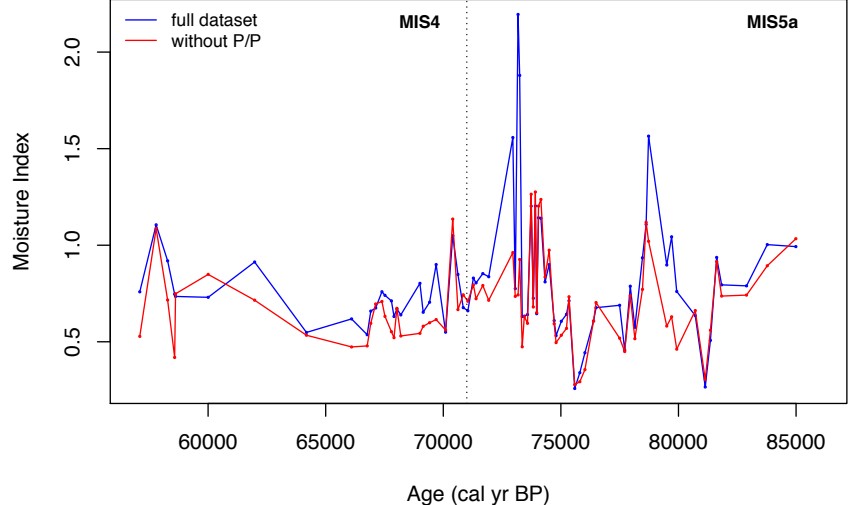






Figure 4: Reconstructed mean temperature of the coldest month (MTCO, °C), growing
degree days above a base level of 0° C (GDD$_0$) and moisture index (the ratio of annual
precipitation to annual potential evapotranspiration, MI). Only samples with a Hill's
N2 biodiversity index >2 are plotted. The Marine Isotope Stages (MIS) and substages
are shown by vertical dotted lines and labelled; we also show the transition interval
between MIS6 and MIS5e. Red dots indicate the modern climate calculated from the
elevation-corrected climate data from the Climate Research Unit (CRU) CL 2.0 data
set.

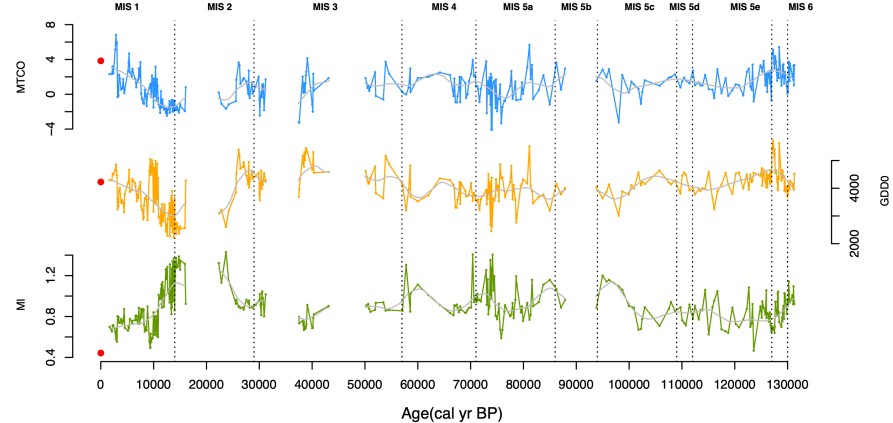






Figure 5: The correlation of the temperature seasonality and insolation. The black line
in the top plot is the normalized difference of reconstructed mean temperature of the
coldest month and the mean temperature of the warmest month calculated based on
MTCO and $GDD_0$ (Appendix 1). The orange line in the top plot is the difference
between July and January insolation in W m$^{-2}$ at 40.49 °N (the latitude of
Villarquemado). The bottom panel shows mid-monthly insolation anomalies
(compared to present) in W m$^{-2}$ at 40.49N through time.

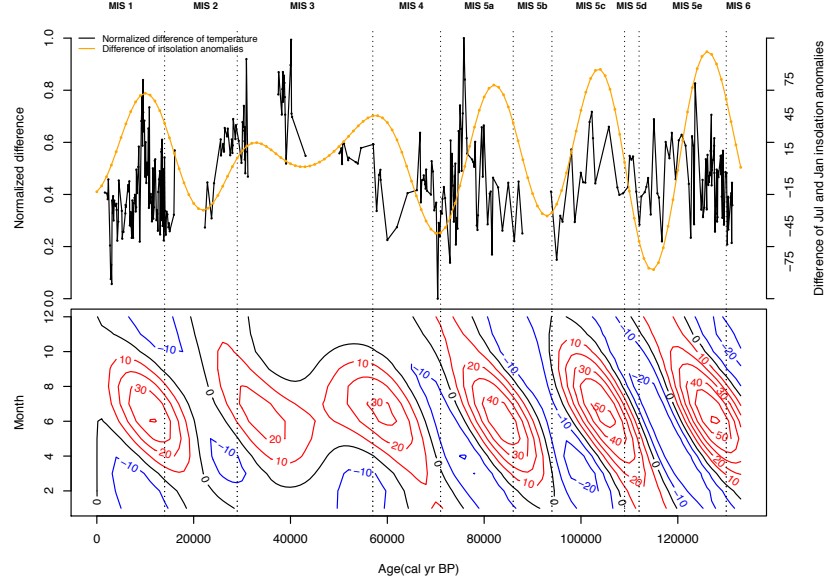






Figure 6: Scatter plot showing the impact of the $[CO_2]$ correction on the reconstructed
moisture index (MI). The coloured dots represent the implied change in the
reconstructions, grouped according to level of the actual $[CO_2]$ at that time (in ppm).

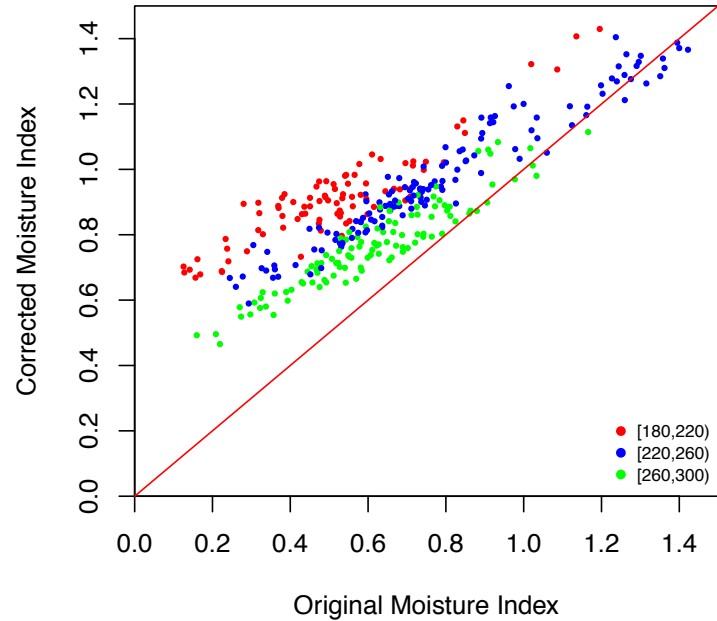




Table 1: Summary statistics for the first three axes of CCA in the whole European data
set. The analysis was based on 6458 sites, 196 taxa and three bioclimatic variables:
mean temperature of the coldest month (MTCO, °C), growing degree days above 0°C
(GDD$_0$) and the moisture index (MI). We also show the summary statistics for the
ANOVA-like permutation test (999 permutations).

| **Axes:** | Axis 1 | Axis 2 | Axis 3 | Variance Inflation Factor |
|---|---|---|---|---|
| Constrained eigenvalues | 0.376 | 0.140 | 0.060 | |
| Cumulative percentage variance of species-environment relationship | 65.4 | 89.7 | 100.0 | |
| Species-environment correlations | 0.83 | 0.61 | 0.47 | |
| Correlations of the environmental variables with the axes: | | | | |
| MI | 0.693 | 0.594 | -0.409 | 2.28 |
| T$_{min}$ | -0.883 | 0.456 | -0.109 | 3.21 |
| GDD$_0$ | -0.946 | -0.193 | -0.260 | 5.23 |

| | Df | ChiSquare | F | Pr (>F) |
|---|---|---|---|---|
| Whole model: | 3 | 0.5757 | 64.393 | 0.001 |
| Bioclimatic variables: | | | | |
| MI | 1 | 0.2399 | 80.506 | 0.001 |
| MTCO | 1 | 0.2543 | 85.344 | 0.001 |
| GDD$_0$ | 1 | 0.0814 | 27.329 | 0.001 |
| Aexs: | | | | |
| CCA 1 | 1 | 0.3763 | 126.253 | 0.001 |
| CCA 2 | 1 | 0.1399 | 46.956 | 0.001 |
| CCA 3 | 1 | 0.0595 | 19.971 | 0.001 |






Table 2: The results of randomisation *t*-test on the leave-one-out cross-validated
predictions of the weighted averaging-partial least squares (WA-PLS) regression
models used for the climate reconstructions. The final model is based on 194 taxa,
omitting Poaceae and Polypdiales.  Selected components in the final model are marked
in bold.

| WA-PLS component | RMSEP | $r^2$ | Maximum bias | p |
|---|---|---|---|---|
| **MTCO** | | | | |
| 1 | 5.308 | 0.624 | 13.551 | 0.001 |
| 2 | 4.967 | 0.671 | 9.044 | 0.001 |
| 3 | 4.852 | 0.686 | 8.919 | 0.001 |
| **4** | **4.829** | **0.689** | **9.882** | **0.035** |
| 5 | 4.840 | 0.688 | 9.811 | 0.729 |
| **GDD$_0$** | | | | |
| 1 | 965.106 | 0.618 | 2529.455 | 0.001 |
| 2 | 909.640 | 0.660 | 2287.800 | 0.001 |
| 3 | 892.668 | 0.673 | 2195.842 | 0.001 |
| **4** | **890.397** | **0.675** | **2278.140** | **0.022** |
| 5 | 891.459 | 0.674 | 2305.115 | 0.774 |
| **MI** | | | | |
| 1 | 0.452 | 0.444 | 3.868 | 0.001 |
| 2 | 0.430 | 0.497 | 3.466 | 0.001 |
| **3** | **0.427** | **0.505** | **3.439** | **0.004** |
| 4 | 0.426 | 0.506 | 3.497 | 0.407 |






Table 3: Reconstructed average values of mean temperature of the coldest month
(MTCO, °C), growing degree days above a base level of 0° C (GDD$_0$) and moisture
index (the ratio of annual precipitation to annual potential evapotranspiration, MI) for
Marine Isotope Stages (MIS) and substages, calculated from the interpolated yearly
values of each variable.

|        | MTCO  | GDD$_0$ | MI   |
|--------|-------|---------|------|
| MIS1   | 0.81  | 3729    | 0.81 |
| MIS2   | -0.22 | 3631    | 1.10 |
| MIS3   | 1.04  | 4542    | 0.85 |
| MIS4   | 1.52  | 3894    | 0.97 |
| MIS5a  | 0.65  | 3817    | 0.96 |
| MIS5b  | 2.08  | 3955    | 0.97 |
| MIS5c  | 1.15  | 4063    | 0.92 |
| MIS5d  | 1.52  | 4168    | 0.82 |
| MIS5e  | 1.25  | 4182    | 0.80 |
| MIS6   | 2.34  | 4102    | 0.99 |
