# Peer review of "Climate changes in interior semi-arid Spain from the last interglacial to the"

_Climate of the Past, 2019_

## Short Comment (SC1) · 20 Feb 2019

This manuscript contains some excellent material (e.g. Appendix 1, correcting estimates of moisture for [CO2]) and much less good material (e.g. pollen-based reconstructions, numerical methods, component selection, SI Table 2, no data for the fossil sequence, no pollen diagram showing major taxa, no information about how the sequence has been constructed from the several pollen diagrams published from the sites, no age-depth model). I am surprised that the statistical significance of the reconstructions was not assessed using the Telford and Birks (2011, QQSR) randomisation procedure. Such an assessment would provide a critical test of how potentially robust the reconstructions are. Overall I recommend major revisions to some parts plus extending the manuscript to be more complete. It is potentially a very valuable study, it

simply needs to be presented more fully so that readers can evaluate the reconstructions, the discussion, the chronology, etc. for themselves.

Some specific comments: Line 41: "long-term" rather than 'longterm'

Line 49: "ice-sheet volume" rather than 'ice sheet volume'

Line 63: Two of these papers (González-Sampériz et al. 2013; Aranbarri et al. 2014) present parts of the entire pollen sequence from Villarquimado (V). It would be a great help to the readers of this manuscript to have a more detailed complete pollen diagram than Figure 2 that only shows four groups of taxa.

Line 66: Valero-Garcés et al. needs a year. Again it would be useful to see the age-depth model for V so that one can judge 'the quality of the age model'.

Lines 102-113: I assume that all the data not already in public databases such as EMPD, EPD, and Pangaea will soon be publicly available (e.g. some of the EMBSeCBIO data that are not in the EPD).

Line 139: Were these GAMs constructed using pollen percentages? If so, what taxa were included in the calculation sum?

Line 148: It would have been useful to present 2–4 of these GAMs.

Line 156: Were spores of pteridophytes and lycophytes included in the pollen sum?

Line 218: Citation needed for the statement about WA-PLs being 'relatively robust to spatial autocorrelation' (e.g. Telford and Birks 2005 or 2008 QSR).

Line 226: ter Braak et al. 1993 is missing in the reference list.

Lines 221-227: I am confused here. Van der Voet's method tests the equality of predictions (reconstructions) from two models. As shown by Juggins and Birks (2012 DPER vol 5, chapter 14), this randomisation t-test provides additional information to help discriminate "hidden" over-fitting from real systematic model improvement, whereas the

selection of WA-PLS components on the basis of RMSEP can lead to the selection of an inappropriate model and lead to a correspondingly over-optimistic impression of the prediction error. Why use the Van der Voet t-test to select the number of components and then use the lowest RMSEP as a basis for selecting WA-PLS components? The Van der Voet test is less prone to lead to over-fitting (see Juggins and Birks 2012).

Line 239: Is there a word missing after 'effects'?

Lines 258-261: The CCA (Table 1 lower half) does not show that the three climatic variables have an independent contribution to explaining variation in pollen abundances as you do not appear to have conditional (partial) CCAs with MI as the predictor variable and MTCO and GDD0 as covariables. Also it should be "Axes" not 'Aexs'. See Juggins (2013 QSR) for a detailed discussion of the major problems of identifying independent predictor variables.

Lines 263-264: Three and four components are quite high – does the Van der Voet test show that these components are significant?

Line 269: But Polypodiales are not really aquatic. Of the taxa you list in S1 Table 2, Thelypteris palustris is the only mire plant. None are aquatics.

Line 480: Is there a word missing after 'regarding'?

Lines 542-554: A corrected list of authors of this paper was published as a correction in VHA. Line 623: "Quaternary" not 'quaternary'

Lines 823-828: Maps a and b are very small and b is impossible to read.

Figure 4: It is a missed opportunity not to have used bootstrapping (available in rioja) to estimate sample-specific errors for these three reconstructions.

Table 2: I am confused. These do not seem to me to be the results of the Van der Voet randomisation t-test. Also here it says that Poaceae and Polypodiales (not correct spelling) were omitted but on line 159 you say 196 taxa were used. What were actually

used – 194 or 196 taxa?

SI Table 2: I am not sure that this is needed when much more relevant basic data are not given, e.g. the fossil pollen counts for the V sequence, modern pollen data not in the EPB, EMPD, or Pangaea (e.g. some of the data in Marinova et al. 2017). A quick perusal of SI Table 2 indicates some errors:

• some of the taxa included in Asteroideae have very distinct pollen (Ambrosia, Xanthium)

• same for Carduoideae, Caryophyllaceae (Spergula, Paronychia, Illecebrum, Agrostemma)

• Succisa is not in the Caryophyllaceae

• Fabaceae and Fabaceae (herbs) is an unsatisfactory division

• Tuberaria pollen is like Helianthemum

• Hepatica and Pulsatilla pollen are like Anemone pollen

• Hypericaceae pollen are the same as Guttiferae = Clusiaceae

• Hippuris (in Plantaginaceae) is an aquatic whose pollen is totally different from Plantago pollen

• Cryptomeria is a tree, not a fern (Pteridaceae), many of the Pteridaceae have distinct spores (e.g. Adiantum, Cryptogramma)

• same for Rosaceae (Fragaria has Potentilla-type pollen)

• some of the taxa in your Scrophulariaceae are now in the Orobanchaceae (Euphrasia, Pedicularis, Rhinanthus).

This is an allocation of pollen types to higher plant taxonomy names, not to similar pollen types that have been given different names by different analysts. It is a record of how you have amalgamated the 249 taxa but there are errors in it.

SI Table 4: It seems a bit excessive to give estimated ages to one decimal place!

SI Table 5: References are missing.

References: Need checking against the citations in the text.

---

## Author Comment (AC1) · 8 Mar 2019

We note that the reviewer has uploaded the same comments twice, although the formats are changed. Since we assume this was an error, we only provide a single response here.

Original comments are in **bold**, our response in *italics*, with suggested changes in the text in normal script.

**Line 41: "long-term" rather than 'longterm'**
*We will change this.*

**Line 49: "ice-sheet volume" rather than 'ice sheet volume'**
*We will change this*

**Line 66: Valero-Garcés et al. needs a year**
*We have added 2019. This paper is accepted for publication.*

**Line 63: Two of these papers (González-Sampériz et al. 2013; Aranbarri et al. 2014) present parts of the entire pollen sequence from Villarquimado (V). It would be a great help to the readers of this manuscript to have a more detailed complete pollen diagram than Figure 2 that only shows four groups of taxa.**

*It is true that the Aranbarri et al. (2014) only covers the last 13,000 years, but González-Sampériz et al. 2013 provide a simplified diagram of part of the whole sequence. A more detailed paper describing the pollen sequence and vegetation history from Villarquemado is currently in preparation. The goal of the current paper is to provide a climate reconstruction rather than to document the pollen record. We therefore showed Poaceae and Polypodiales in order to illustrate the spiky nature of the record –which affects our reconstructions. Similarly, we showed the biodiversity index because we use this to explain why some samples do not provide good reconstructions. The major changes of climate are reflected in the changes between Mediterranean and Steppe plant functional type, which is why we show these. However, we will expand Figure 2 to include more key taxa groups and provide more comprehensive information regarding the palynological sequence.*

**Line 66: Again it would be useful to see the age depth model for V so that one can judge 'the quality of the age model'.**
*The Valero-Garcés et al., which describes the construction of the age-depth model in great detail is accepted for publication and is expected to be available within the next few weeks. However, we will include the Bayesian age model for the sequence, derived from this paper, in our SI.*

**Lines 102-113: I assume that all the data not already in public databases such as EMPD, EPD, and Pangaea will soon be publicly available (e.g. some of the EMBSeCBIO data that are not in the EPD).**
*The modern surface pollen data set that we use for these reconstructions is publically available and we will give the DOI for the data set in the text. The fossil data will be made available via Neotoma and/or EPD upon publication of the paper currently in preparation describing the palynological sequence in detail.*

**Line 139: Were these GAMs constructed using pollen percentages? If so, what**

**taxa were included in the calculation sum?**
*We will rewrite this text as follows:*
The GAMs were implemented with the mgcv R package (Wood, 2017). The R implementation makes the selection of the smoothing parameters automatic (Guisan et al., 2002). We used pollen percentages based on a pollen sum that included all of the 196 taxa.

**Line 148: It would have been useful to present 2–4 of these GAMs.**
*Since these are large figures, we will include two examples of the GAMs in the Supplementary Material (Supplementary Figure 1 and 2). We will also change the figure numbering in the text.*

**Line 156: Were spores of pteridophytes and lycophytes included in the pollen sum?**
*The pollen sum includes all of the taxa that were used for the reconstruction, please see Supplementary Table 2.*

**Line 218: Citation needed for the statement about WA-PLs being 'relatively robust to spatial autocorrelation' (e.g. Telford and Birks 2005 or 2008 QSR).**
*We will add a reference here.*

**Line 226: ter Braak et al. 1993 is missing in the reference list**
*Thanks for pointing this out. We will add it into the reference list.*

**Lines 221-227: Why use the Van der Voet t-test to select the number of components and then use the lowest RMSEP as a basis for selecting WA-PLS components? The Van der Voet test is less prone to lead to over-fitting (see Juggins and Birks 2012).**
*The selected component was first determined on the basis that (a) the Van der Voet test showed it was significant, whereas subsequent components were non-significant and then (b) by choosing the lowest RMSEP of the significant components. We obviously did not explain this clearly enough and will rewrite the text as follows:*

The performance of the calibration models was assessed through leave one-out cross validation. The number of components used in each model was estimated through a randomisation t-test on the results of this cross validation (Van der Voet, 1994). We selected the significant component with the lowest root mean square error of prediction (RMSEP), but only if here was a significant improvement in RMSEP relative to a lower number of components – since including more components can result in over-fitting of the data so that model predictive value decreases (ter Braak et al., 1993). We checked that the final transfer functions had a high R2 for prediction and a low maximum bias.

**Line 239: Is there a word missing after 'effects'?**
*This should have read "effects of" and we will correct this*

**Lines 258-261: The CCA (Table 1 lower half) does not show that the three climatic variables have an independent contribution to explaining variation in pollen abundances as you do not appear to have conditional (partial) CCAs with MI as the predictor variable and MTCO and GDD0 as covariables. Also it should be "Axes" not 'Aexs'.**

*We have corrected the typo.*
*We have run a partial CCA as suggested, with each climate variable as the predictor and the others as covariables. This shows that they all have an independent and significant (p < 0.001) contribution to explaining the variability. We will add this information in the paper.*

**Lines 263-264: Three and four components are quite high – does the Van der Voet test show that these components are significant?**
*As the P values in Table 2 show, these components are all significant at the 95% confidence level. We have also checked the impact of using component 3 for MTCO and GDD, and component 2 for MI (i.e. the last components that are significant at 99% confidence rather than 95% confidence), and found that this does not affect the reconstructions noticeably or change the conclusions of the paper. We will include these alternative reconstructions in the SI.*

**Line 269: But Polypodiales are not really aquatic. Of the taxa you list in S1 Table 2, Thelypteris palustris is the only mire plant. None are aquatics.**
*We have expressed this badly. We think that the distinct Polypodiales peak values are the result of inwash into this semi-aquatic environment (as shown by the nature of the sedimentary record) because the spores are persistent in the environment. We will rephrase this as follows:*
These correspond to samples that have unusually high values of either Poaceae or Polypodiales (Fig. 2), and where the sedimentary record indicates that environmental conditions were fluctuating and the basin was occupied by wetlands. Under these conditions, the anomalously high peaks of Poaceae likely correspond to reeds and the anomalously high peaks of Polypodiales could represent inwash.

**Line 480: Is there a word missing after 'regarding'?**
*This should have read "regarding the" and we will correct this.*

**Lines 542-554: A corrected list of authors of this paper was published as a correction in VHA.**
*We will indicate that there has been a correction to the paper in the reference list.*

**Line 623: "Quaternary" not 'quaternary'**
*We will correct this*

**Lines 823-828: Maps a and b are very small and b is impossible to read.**
*We agree that these two maps are very small. Figure a is not necessary for this paper and we will remove it and we will redraw Figure c to make the location of Villarquemado clearer. We think that Figure b is helpful in understanding the setting of the site. We will therefore preserve this figure but treat it as a separate figure from plot c. We will renumber the figures in the main text accordingly.*

**Figure 4: It is a missed opportunity not to have used bootstrapping (available in rioja) to estimate sample-specific errors for these three reconstructions.**
*The bootstrapped errors as calculated in rioja are large (MTCO = 4.8°C, GDD = 900, MI =0.4) but these values are comparable to other WA-PLS reconstructions (e.g. Lake Ohrid, MAT = 5°C, PANN = 200 mm, see Sinapoli et al., 2019). The issue here is that the standard leave-one-out bootstrapping approach provides a measure of the*

*stability of the model to the individual samples in the training data set. We have applied an alternative approach of creating 1000 training data sets by random selection and then examining the stability of the taxon coefficients. We then apply these different models to estimate the reconstruction error associated with individual samples in the fossil record. In this approach, samples which are dominated by taxa with stable coefficients will have relatively narrow confidence intervals while samples dominated by taxa with non-stable coefficients will yield large errors. We believe this approach gives more realistic confidence intervals for individual samples than the standard method. We will update the figures to include these confidence intervals.*

**Table 2: These do not seem to me to be the results of the Van der Voet randomisation t-test. Also here it says that Poaceae and Polypodiales (not correct spelling) were omitted but on line 159 you say 196 taxa were used. What were actually used – 194 or 196 taxa?**
*The P values are indeed from the Van der Voet randomization t-test. We will modify the caption to make this clear as follows:*
The results of the leave-one-out cross-validated predictions of the weighted averaging-partial least squares (WA-PLS) regression models used for the climate reconstructions. The P values are derived from the randomisation t-test on these results. The final model is based on 194 taxa, omitting Poaceae and Polypodiales. Selected components in the final model are marked in bold.

*We originally used 196 taxa to do the reconstruction but, as explained in the results, we found that Poaceae and Polypodiales had highly anomalous peaks that influenced the reconstructions and we therefore left them out in the final reconstruction. As we point out, this influenced the reconstructions for the samples with anomalously high Poaceae and Polypodiales but did not affect the reconstructions for the other samples. Nevertheless, as we present results both the reconstructions with and without Poaceae and Polypodiales (Supplementary Table 3) we need to explain which results are based on 196 taxa and which are based on 194 taxa. We will modify the description in the methods (Line 158-159) to read:*
Amalgamated taxa that occur in less than 10 sites were not considered in the analyses, reducing the number examined from 249 to 196 taxa. We also ran further analyses after removing two taxa that displayed anomalous behavior. The final results are therefore based on 194 taxa.

*We will also modify the text at Line 270-272 to read:*
Both Poaceae and Polypodiales were therefore removed from the final WA-PLS model (Table 2), reducing the total number of taxa considered from 196 to 194.

**SI Table 2: I am not sure that this is needed when much more relevant basic data are not given.**
*As explained above, basic information on the age model and the pollen record are given in other publications. We will include a figure in the SI with the age-depth model and we will expand the presentation of the summary pollen diagram to include more functional types. However, our focus here is on the climate reconstruction, and we think it is useful to have this table to explain how taxa were amalgamated to make the reconstruction.*
*It is true that some pollen types that we have grouped together are morphological distinct. However, our data set was compiled from multiple different published data*

*sets and not every palynologist made these distinctions. Some morphologically distinct pollen types are only recorded at a very small number of sites and often these sites are geographically clustered, suggesting a systematic recording bias. Including taxa that are only recorded rarely (or show geographic clustering suggesting that they have not been sampled across the whole of their potential climate range) would lead to misleading reconstructions. Niche conservatism would suggest that higher taxa should have a coherent environmental distribution (and this is more reasonable than the idea that a morphologically distinctive pollen type would have a coherent distribution). Furthermore, we checked the climate space of component taxa that were amalgamated into higher taxa using the GAMs to ensure that the grouped taxa showed similar climatic preferences.*

a) Asteroideae have very distinct pollen (Ambrosia, Xanthium). *Yes, but Ambrosia is only identified <3% of the sites and Xanthium only <5% of the sites in the data set. According to the GAMs, they occupy the same climate niche*
b) Carduoideae, Caryophyllaceae (Spergula, Paronychia, Illecebrum, Agrostemma). *The same is true of these species: Spergula (<2%), Paronychia (<0.1%), Illecebrum (<0.1%) Agrostemma (<0.1%)*
c) Succisa is not in the Caryophyllaceae. *Succisa is in the Caprifoliaceae and this was a mistake in our Table and we will correct this.*
d) Fabaceae and Fabaceae (herbs) is an unsatisfactory division. *We agree this is unsatisfactory. However, at a large number of sites herbaceous taxa have been identified separately from Fabaceae as a whole. The identified herbaceaous taxa are usually rare and would not be used unless they were grouped together, and we checked they occupied a similar area of climate space and that this was distinctive from the climate space occupied by the taxon that was simply called Fabaceae in the same samples.*
e) Tuberaria pollen is like Helianthemum. *Helianthemum is identified very commonly in the data set (12% of the sites), whereas Tuberaria is only found at 0.03% of the sites. We could also have included Helianthemum in the Cistaceae, but this seemed like a loss of information given that many palynologists have identified it explicitly.*
f) Hepatica and Pulsatilla pollen are like Anemone pollen. *It is true that these are rare and we have amalgamated them into Ranunculaceae and re-run the reconstructions. The change has no effect on the results.*
g) Hypericaceae pollen are the same as Guttiferae = Clusiaceae.
*Hypericaceae is an accepted family and Hypericum is explicitly identified at 252 sites in the data base; Clusiaceae (on the other hand) is a separate, regonised family but representatives are extremely rare in the data set and therefore not used in our reconstructions.*
h) Hippuris (in Plantaginaceae) is an aquatic whose pollen is totally different from Plantago pollen. *We agree that Hippuris, which we have put into the Plantaginaceae, is an aquatic and should be omitted from the data set. It only occurs in a few sites and omitting this does not affect our reconstructions.*
i) Cryptomeria is a tree, not a fern (Pteridaceae), many of the Pteridaceae have distinct spores (e.g. Adiantum, Cryptogramma). *The inclusion of Cryptomeria into the Pteridaceae was a mistake in the table and we will correct this. It only occurs in one site in the data set, and so will not be used in the reconstructions.*
j) Rosaceae (Fragaria has Potentilla-type pollen). *Fragaria only occur at 5 sites in the data set but Potentilla occurs 906 times and furthermore occupies a distinctive climate space. We could also have included Potentilla in the Rosaceae, but this seemed like a loss of information given that many palynologists have identified it explicitly.*

k) some of the taxa in your Scrophulariaceae are now in the Orobanchaceae (Euphrasia, Pedicularis, Rhinanthus). *Yes these three genera are now in the Orobanchaceae, as in fact is Melampyrum – which we also included in Scrophulariaceae. We will remove these from Scrophulariaceae and include Orobancaceae as a new taxon. They only occur at a few sites and this change has no impact on the reconstructions*.

---

## Short Comment (SC2) · 17 Mar 2019

Please note that the two sets of comments posted by John Birks are the same (except for formatting)

---

## Author Comment (AC2) · 4 Apr 2019

We thank the referee for their positive comments about the value of this paper, and particularly the application of a correction for the physiological effects of low CO2 and the use of a more robust technique and expanded training data set for the reconstructions.

In response to the specific comments (Original comments are in **bold**, our response in *italics*, with suggested changes in the text in normal script):

**Age control and stratigraphy – the referee suggests that we could discuss further, and help the reader to assess, the chronological uncertainties and consider the stratigraphical complexity of this very dynamic environmental setting which must also bear on the uncertainty of the age-model, and that comments about very specific D-O events (e.g. lines 331-334, line 404) should be checked and revised to reflect realistic caution about the attributions.**

*We agree that the quality of the age model is very important. We cite the paper describing the age model in detail in the text (Valero-Garcés et al.) – this paper is accepted for publication and we expect this to be published in the next few weeks. However, we intend to include a new figure with the Bayesian age model for this record and some explanation in the Supplementary Information, so that the readers can make an assessment of the reliability. We identified the D-O events based on the known age of these events in the Greenland ice core, and we made no adjustment to our age model. The fact that pronounced changes occur at the appropriate age may, of course, be by chance but this seems unlikely. However, we can modify the text to make it clear that these were identified purely based on their ages (line 331 onward) as follows:*

Some of these (Supplementary Fig. 4) clearly occur at times that correspond to D-O events, including D-O 20 (72.28-70.28 cal ka) and D-O 19 (76.4-74 cal ka) in MIS5a and D-O 9 (40.11-39.81 cal ka) and D-O 8 (38.17-36.57 cal ka) in MIS3. Heinrich Stadial 2 (26.45-24.25 cal ka) also clearly corresponds to an interval of year-round cooling in our reconstructions. Gaps in the pollen record, and poor dating resolution in some parts of the record, preclude identification of all of the D-O and Heinrich events. However, where abrupt events that appear to correspond to D-O events are registered, they were characterized by a marked increase in seasonality – this explains the apparently anomalous high seasonality recorded during some parts of the glacial (Fig. 5).

*We will also modify the text at line 404 to be more specific, as follows:*

Orbital forcing was not the only cause of enhanced seasonality at Villarquemado, since we also see enhanced seasonality associated with intervals of abrupt warming that appear to correlate with D-O events (e.g. D-O 9).

**Given that forest development in dry sectors of the Mediterranean is generally considered to be limited by moisture availability, the authors should comment on why the lowest moisture availability is reconstructed for the interval with highest Mediterranean taxa and why the vegetation transition from xerophytic grassland steppe to forest development would imply reduced moisture availability.**

*The highest levels of MI occur in the late glacial when the abundance of Mediterranean taxa are low, but this probably reflects the impact of CO2 on tree growth. Relatively high lake levels, and higher moisture availability during the Lateglacial than the first part of the Holocene, have been recorded in other regional lacustrine sequences (i.e., Valero-Garcés et al., 2000, 2004; González-Sampériz et al., 2006, 2017; Morellón et al., 2009). Low MI around 10ka corresponds to an interval when steppe is prominent (increase in Artemisia and Chenopodiaceae:*

*Aranbarri et al., 2014) and there is a marked decrease in Pinus diploxylon (and Cyperaceae). These arid and/or still cool conditions for the beginning of the Holocene have been identified in different lacustrine sequences from inner Iberia and related to intense evapotranspiration due to high summer insolation and extreme warm temperatures (Morellón et al., 2018). There is a slight increase in MI during the middle Holocene when seasonality is reduced and therefore, evapotranspiration too; this appears to correspond to an increase in Mediterranean taxa (mainly evergreen Quercus) (Aranbarri et al., 2014). In addition MI is a ratio of annual precipitation to evapotranspiation; our Mediterranean assemblage includes trees and shrubs that are tolerant to increasing evapotranspiration (Quercus rotundifolia, Oleaceae, Rhamnus, Helianthemum) which would be a likely scenario during the Lateglacial-Early Holocene with increasing temperature. Such situation with increasing temperature, and evapotranspiration under increasing $CO_2$ is fully compatible with the development of woody Mediterranean vegetation.*

*- Aranbarri, J., González-Sampériz, P., Valero-Garcés, B., Moreno, A., Gil-Romera, G., Sevilla-Callejo, M., García-Prieto, E., Di Rita, F., Mata, M. P., Morellón, M., Magri, D., Rodríguez-Lazaro, J., and Carrion, J. S.: Rapid climatic changes and resilient vegetation during the Lateglacial and Holocene in a continental region of south-western Europe, Global Planet. Change, 114, 50-65,*
*- González-Sampériz, P., Valero-Garcés, B.L., Moreno, A., Morellón, M., Navas, A., Machín, J., Delgado-Huertas, A., 2008. Vegetation changes and hydrological fluctuations in the Central Ebro Basin (NE Spain) since the Late Glacial period: Saline lake records. Palaeogeogr. Palaeoclimatol. Palaeoecol. 259, 157–181.*
*- González-Sampériz, P., Aranbarri, J., Pérez-Sanz, A., Gil-Romera, G., Moreno, A., Leunda, M., Sevilla-Callejo, M., Corella, J.P., Morellón, M., Oliva, B., Valero-Garcés, B., 2017. Environmental and climate change in the southern Central Pyrenees since the last glacial maximum: a view from the lake records. Catena 149 (vol.3), 668–688.*
*- Morellón, M., Valero-Garcés, B., Vegas-Vilarrúbia, T., González-Sampériz, P., Romero, Ó., Delgado-Huertas, A., Mata, P., Moreno, A., Rico, M., Corella, J.P., 2009. Lateglacial and Holocene palaeohydrology in the western Mediterranean region: the Lake Estanya record (NE Spain). Quat. Sci. Rev. 28, 2582–2599.*
*- Morellón, M., Aranbarri J, Moreno A González-Sampériz, P., Valero-Garcés, B.L. 2018. Early Holocene humidity patterns in the Iberian Peninsula reconstructed from lake, pollen and speleothem records. Quaternary Science Reviews 18: 1-18.*
*- Valero-Garcés, B.L., González-Sampériz, P., Delgado-Huertas, A., Navas, A., Machín, J., Kelts, K., 2000b. Lateglacial and Late Holocene environmental and vegetational change in Salada Mediana, central Ebro Basin, Spain. Quat. Int. 73–74, 29–46.*
*- Valero-Garcés, B.L., González-Sampériz, P., Navas, A., Machín, J., Delgado-Huertas, A., Peña-Monné, J.L., Sancho-Marcén, C., Stevenson, T., Davis, B., 2004. Paleohydrological fluctuations and steppe vegetation during the last glacial maximum in the central Ebro valley (NE Spain). Quat. Int. 122, 43–55.*

**Supplementary Figure 3 showing the impact of moisture index correction for CO2 against time is one of the major findings of the study, and could usefully be incorporated in the main paper.**
*We agree that this is an interesting figure and we will move it into the main text.*

Minor comments:

**Line 61. The record cannot really be described as "continuous" in light of several intervals with poor pollen preservation and hence no reported pollen spectra (e.g. lines 207-208).**
*We agree that there are gaps in the pollen record, and will take out the word continuous.*

**Line 197. Is the local evergreen oak Q. ilex subsp ilex or Q. ilex subsp rotundifolia?**
*The local evergreen oak today is Q. ilex subsp rotundifolia. Unfortunately, it is not possible to distinguish these sub-species in the pollen record. However it is highly unlikely that Q. ilex subsp ilex was present in the area as this taxon would not tolerate the extreme temperatures that occurred in Villarquemado throughout the whole sequence because of its continentality.*

**Line 197. Q. faginea is generally classed as a deciduous (or marcescent) species, rather than evergreen.**
*It is somewhat difficult to know whether to classify marcescent species, such as Q. faginea, as deciduous or evergreen. For our reconstructions (see Supplement) we divide the Quercus taxa into the three groups, strictly evergreen, strictly deciduous and an intermediate class which includes marcescent species. We will modify the text to read:*
 **….. by evergreen or marcescent trees**

**Line 207-208. Round ages to nearest 10 or 100 years.**
*We agree that we were being over-precise here and will round these ages to the nearest 100 years. The revised text will read:*
There are intervals with poor pollen preservation between ca 16 000 and 22 300, 31 200 and 37 500, 43 100 and 50 100, and 87 900 and 93 800 cal yr BP.

**Line 423-424. This statement seems a bit too strong – the recent work at Padul provides a pollen record that is arguably more complete (i.e. not containing pollen hiatuses), and could ultimately provide a valuable comparison data set for climate reconstruction**
*We cite the earlier Camuera et al. (2018) paper (lines 69) but had not seen the later paper (or its corrigendum). The Padul record may be more complete and have no hiatuses, but we still feel (as we said in the original text, lines 70-72) that the age model is less well-constrained than that of Villarquemado. Specifically, although the upper 30 m is very well-dated using 42 AMS dates, the lower part of the core is only dated by linear extrapolation of sedimentation rates. Nevertheless, we agree that this is a good record and that it would be worthwhile to try and reconstruct the climate at this site for comparison with Villarquemada as soon as the pollen data are made public. We will add the newer Camuera et al references to the text (line 69) and we will modify our statement at lines 423-424 to read:*
provides a relatively complete and well-dated record from continental Iberia

*Camuera, J., Jiménez-Moreno, G., Ramos-Román, M.J., García-Alix, A., Toney, J.L., Anderson, R.S., Jiménez-Espejo, F., Bright, J., Webster, C., Yanes, Y. and Carrión, J.S., 2019. Vegetation and climate changes during the last two glacial-interglacial cycles in the western Mediterranean: A new long pollen record from Padul (southern Iberian Peninsula). Quaternary Science Reviews, 205, pp.86-105.*